# Myc plays an important role in *Drosophila* P-M hybrid dysgenesis to eliminate germline cells with genetic damage

Ryoma Ota [1] & Satoru Kobayashi[1,2 ✉]

Genetic damage in the germline induced by *P*-element mobilization causes a syndrome known as P-M hybrid dysgenesis (HD), which manifests as elevated mutation frequency and loss of germline cells. In this study, we found that Myc plays an important role in eliminating germline cells in the context of HD. *P*-element mobilization resulted in downregulation of Myc expression in the germline. *Myc* knockdown caused germline elimination; conversely, *Myc* overexpression rescued the germline loss caused by *P*-element mobilization. Moreover, restoration of fertility by *Myc* resulted in the production of gametes with elevated mutation frequency and reduced ability to undergo development. Our results demonstrate that *Myc* downregulation mediates elimination of germline cells with accumulated genetic damage, and that failure to remove these cells results in increased production of aberrant gametes. Therefore, we propose that elimination of germline cells mediated by *Myc* downregulation is a quality control mechanism that maintains the genomic integrity of the germline.

[1] Life Science Center for Survival Dynamics, Tsukuba Advanced Research Alliance (TARA), University of Tsukuba, Tsukuba, Ibaraki 305-8577, Japan. [2] Graduate School of Life and Environmental Sciences, University of Tsukuba, Tsukuba, Ibaraki 305-8572, Japan. ✉email: skob@tara.tsukuba.ac.jp

The germline is the only cell lineage that transmits genetic material to the next generation. Genetic damage in the germline causes developmental defects and genetic diseases in progeny. Therefore, suppression of genetic damage and elimination of aberrant germline cells are thought to play a critical role in maintaining the germline integrity of a species[1–4]. Mobilization of transposable elements, a leading cause of genetic damage in the germline, is suppressed by the Piwi-interacting RNA pathway at the transcriptional and posttranscriptional levels[5–12]. However, little evidence exists to support the germline elimination caused by genetic damage. To address this issue, we induced the elimination of germline cells by mobilizing *P*-elements and identified genes rescuing the elimination. We predicted that failure to remove germline cells harboring genetic damage by altering the functions of these genes would increase production of aberrant gametes.

In *Drosophila*, P-M hybrid dysgenesis (HD) is a sterility syndrome attributed to *P*-element mobilization[13,14]. When females lacking *P*-elements (referred to as the M strain) are crossed to males carrying *P*-elements (referred to as the P strain), mobilization of *P*-elements occurs in the germline of their progeny, which in turn exhibit a germline-loss phenotype[13–16]. This germline-loss phenotype is thought to be caused by DNA damage associated with *P*-element mobilization[17]. In ovarian germline cells, DNA damage induces cell-cycle arrest and apoptosis via the functions of two DNA damage response (DDR) genes, *loki* (also known as *mnk*) encoding checkpoint kinase 2 (Chk2) and *p53*[18–21]. In early oogenesis, *loki*/Chk2 is required for cell-cycle arrest of the germline stem cells (GSCs) and their daughters in response to DNA damage[18,20]. By contrast, *p53* is required for cell-cycle re-entry of GSCs and their maintenance[19,20]. Later, in meiotic cells, *p53* acts as a downstream effector of *loki*/Chk2 to induce apoptosis[18,21]. Consistent with their roles in the DNA damage response, *loki*/Chk2 and *p53* also act as modifiers of HD-caused germline loss during oogenesis[22]. Although there is no known link between DNA damage response and *Bruno*, a regulator of oogenesis[23–25], Bruno activity is also required for the GSC loss phenotype in adult HD females[26]. By contrast, at the preadult stage, the roles of *loki*/Chk2 and *p53* in the germline-loss phenotype caused by HD remain elusive; nevertheless, germline loss is observed from embryonic stage 16 onward[16,27,28]. Thus, the mechanisms underlying germline elimination in HD are not fully understood.

Over the course of our experiments to screen transcription factors for germline development and its proliferation, we came across the phenotype caused by knockdown of *Myc*, a basic helix–loop–helix transcription factor[29–31], that was similar to the germline-loss phenotype observed in HD progeny. In this study, we show that *P*-element mobilization caused downregulation of Myc in the germline. Reduction of *Myc* expression in the germline caused the germline-loss phenotype; conversely, overexpression of *Myc* restored fertility impaired by HD, resulting in production of gametes with elevated mutation frequency and reduced ability to develop into offspring. These observations strongly suggest that elimination of germline cells mediated by *Myc* downregulation plays an important role in maintaining the genomic integrity of the germline.

## Results

**Phenotypes caused by HD and *Myc* knockdown.** M-strain females mated with P-strain males produced HD progeny that exhibited a germline-loss phenotype[14,15] (Fig. 1a–d, g). To determine the developmental stages when germline cells were eliminated, we counted the numbers of germline cells in the gonads of HD progeny. In light of the sex difference in germline cell number within a gonad, we examined males and females separately. In males, the average number of germline cells in each gonad of HD progeny embryos was almost identical to that in control embryos (Fig. 1h). However, the number began to decrease in HD progeny at the first instar, and remained low at the second instar (Fig. 1h). In females, the number of germline cells in HD progeny was equivalent to that in control animals until the first instar (Fig. 1i). The number was severely reduced in HD progeny at the third instar, although a subtle reduction of the number in HD progeny was evident at the second instar (Fig. 1i). Thus, severe reduction in the number of germline cells occurred in HD progeny at the first and third instar stages in males and females, respectively, under the conditions we used.

Over the course of our examination of the knockdown phenotypes of transcription factors, we observed that *Myc* knockdown caused a reduction in the number of germline cells similar to that observed in HD progeny. *Myc*, a transcription factor that regulates several cellular processes, including cell proliferation, growth, and death[29–31], was knocked down by expressing a specific shRNA in the germline under the control of *nos-Gal4-VP16* (*nos-Gal4*) driver from embryonic stage 9 onward[32]. When *Myc* was knocked down in the germline, the number of germline cells began to be reduced at the first and third instar stages in males and females, respectively (Fig. 2a, b). Similar results were obtained using another shRNA targeting a distinct region of the Myc mRNA (Supplementary Fig. 1a, b). Furthermore, *Myc* knockdown resulted in the germline-loss phenotype in adult ovaries and testes (Fig. 2c–g). Thus, both HD and *Myc* knockdown decreased germline cell number at the similar stage of development.

These observations led us to speculate that HD decreases Myc expression in the germline. To monitor Myc expression, we used green fluorescent protein (GFP)-tagged Myc protein (Myc-GFP) expressed under the control of *Myc* promoter, which mimics endogenous Myc expression[33–35]. In male germline, Myc-GFP was observed at embryonic stage 15, and remained detectable until spermatogenesis (Supplementary Fig. 2a, c, e, g, i, k). By contrast, in female germline, Myc-GFP was detected at embryonic stage 15, but was indiscernible at the first and early-second instar (Supplementary Fig. 2b, d, f). The signal reappeared in female germline of mid-second instar larvae, and remained detectable until oogenesis (Supplementary Fig. 2h, j, l). In HD progeny, Myc-GFP expression was significantly reduced in the nuclei of germline cells at embryonic stage 17 and the mid-second instar in males and females, respectively (Fig. 2h–m). Furthermore, the levels of Myc-GFP and endogenous Myc mRNAs were significantly reduced in the germline cells of HD males at embryonic stage 17 (Supplementary Fig. 3a–e, g), although we could not detect these mRNAs in the female germline cells at the mid-second instar, presumably because it was difficult for the probes to permeate the gonads. These results suggest that *Myc* expression was reduced in germline cells of HD progeny before germline number severely decreased.

***Myc* overexpression rescues the germline-loss phenotype in HD.** The observations described above suggest that HD causes *Myc* downregulation in the germline, which in turn leads to the germline-loss phenotype. To address this possibility, we overexpressed *Myc* in the germline under the control of *nos-Gal4*. Although UASt is not active in the germline during oogenesis[36,37], *UASt-EGFP* was activated by *nos-Gal4* to produce EGFP protein in the germline of testes at the embryonic stage 17, and in ovaries at the mid-second instar, when Myc expression was reduced by HD (Supplementary Fig. 4). Using *UASt-Myc* and *nos-Gal4*, we overexpressed *Myc* in the germline cells (Supplementary Fig. 3f, g). When *Myc* was overexpressed in the germline

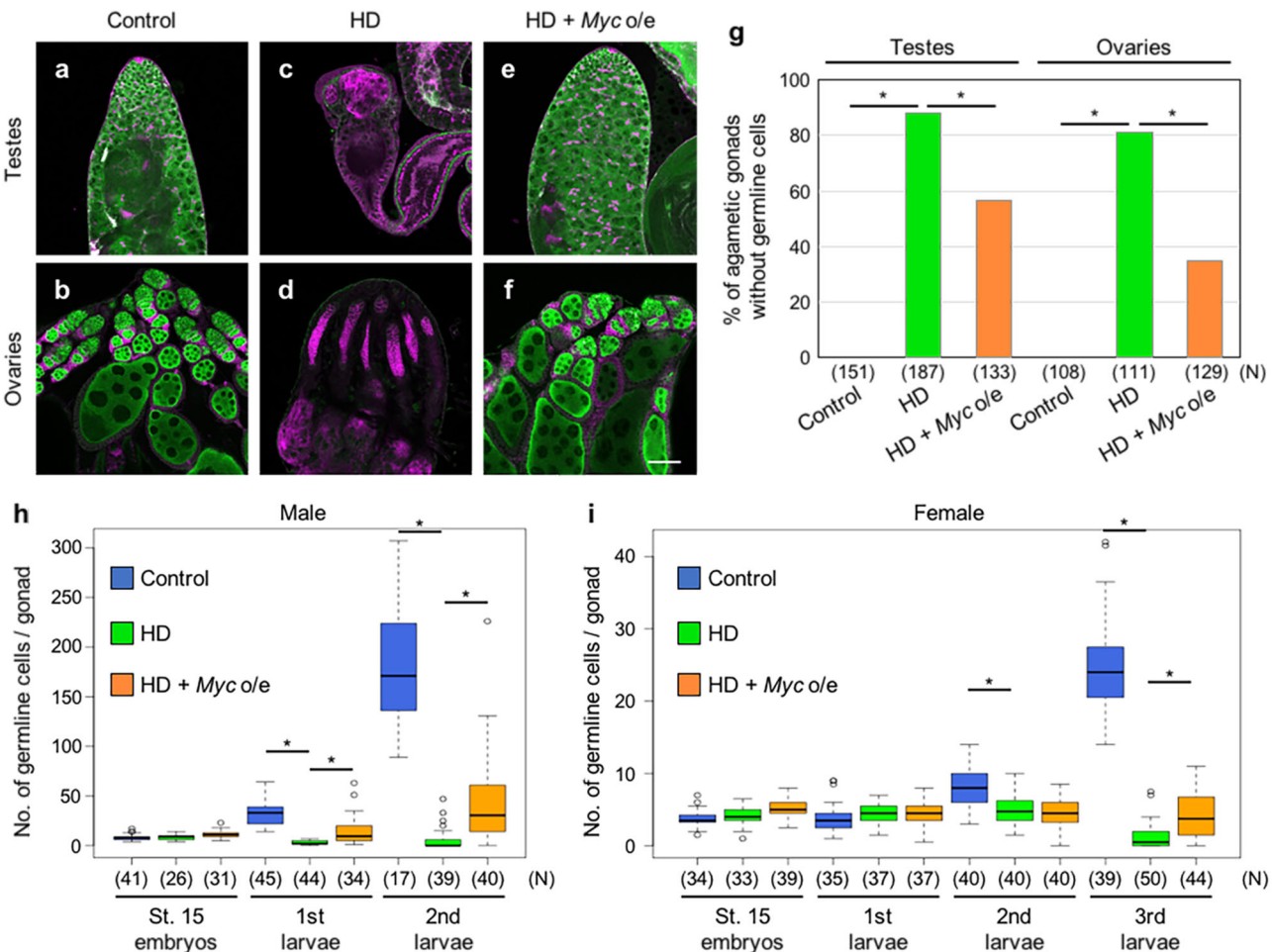

**Fig. 1 *Myc* overexpression rescues the germline-loss phenotype caused by HD.** Testes (**a**, **c**, **e**) and ovaries (**b**, **d**, **f**) of progeny derived from *nos-Gal4/nos-Gal4* females mated with *nos-Gal4/nos-Gal4* (control; **a**, **b**), *nos-Gal4/nos-Gal4* females mated with *Harwich* males (HD; **c**, **d**), and *nos-GAL4 UASt-Myc/TM3, Ser Sb* females mated with *Harwich* males (non-*Ser* and non-*Sb* progeny were selected as HD + *Myc* o/e; **e**, **f**). Gonads were obtained from adults 3–5 days after eclosion and stained for Vasa [a marker for the germline cells (green)], Hts [a marker for spectrosomes and fusomes (magenta)], and Fasciclin III [marker for a subset of somatic cells in germaria and follicles (magenta)]. Scale bar: 50 μm. **g** Percentage of agametic gonads without germline cells in control, HD, and HD + *Myc*-overexpressing (o/e) progeny. Vasa-positive germline cells in gonads were counted. Significance was calculated by two-sided Fisher's exact test; *$P < 0.05$. N: total number of the observed gonads. Similar results were obtained from two biologically independent experiments. The number of germline cells of male (**h**) and female (**i**) progeny derived from *nos-Gal4/nos-Gal4* females mated with *nos-Gal4/nos-Gal4* males (control; blue), *nos-Gal4/nos-Gal4* females mated with *Harwich* males (HD; green), and *nos-GAL4 UASt-Myc/TM3, Act5C-GFP* females mated with *Harwich* males (GFP-negative progeny were selected as HD + *Myc* o/e; orange) at early embryonic stage 15 and early-first, second, and third instar stages. Gonads were stained for Vasa, and Vasa-positive germline cells in gonads were counted. Each box plot represents median value and first and third quartile values. Error bars represent minimum and maximum values. White circles represent outliers. Significance was calculated by two-sided Student's *t* test; *$P < 0.05$. N: total number of the observed gonads. Similar results were obtained from two biologically independent experiments.

cells of HD progeny (HD germline), the percentages of adult testes and ovaries without germline cells were significantly reduced relative to those in HD progeny (Fig. 1g). Conversely, the percentages of normally developed gonads were significantly higher than that in HD progeny (Fig. 1e, f and Supplementary Fig. 5). These observations indicate that *Myc* overexpression in germline cells rescues the germline-loss phenotype caused by HD.

Moreover, we found that *Myc* overexpression rescued the reduction in the number of germline cells at the first and third instar stages in HD males and females, respectively (Fig. 1h, i). Considering that both HD and *Myc* knockdown caused a decrease in the number of germline cells at these stages, and that *Myc* expression was reduced in HD progeny prior to these stages, it is reasonable to conclude that overexpression of *Myc* overrides *Myc* downregulation in the HD germline prior to these stages, which

would otherwise decrease the number of germline cells and consequently the germline-loss phenotype.

**Myc overexpression in HD germline produces aberrant gametes.** Based on the hypothesis that *Myc*-downregulation-dependent reduction in the number of germline cells (MDRG) is the mechanism responsible for eliminating germline cells whose genomes have been damaged by *P*-element mobilization, we predicted that inhibition of MDRG by *Myc* overexpression in the HD germline would result in production of aberrant gametes with a high mutation frequency and a low capacity to produce progeny.

To monitor mutation frequency, we used the *singed-weak* (*sn^w*) allele, which exhibits a weak *sn* phenotype due to the insertion of

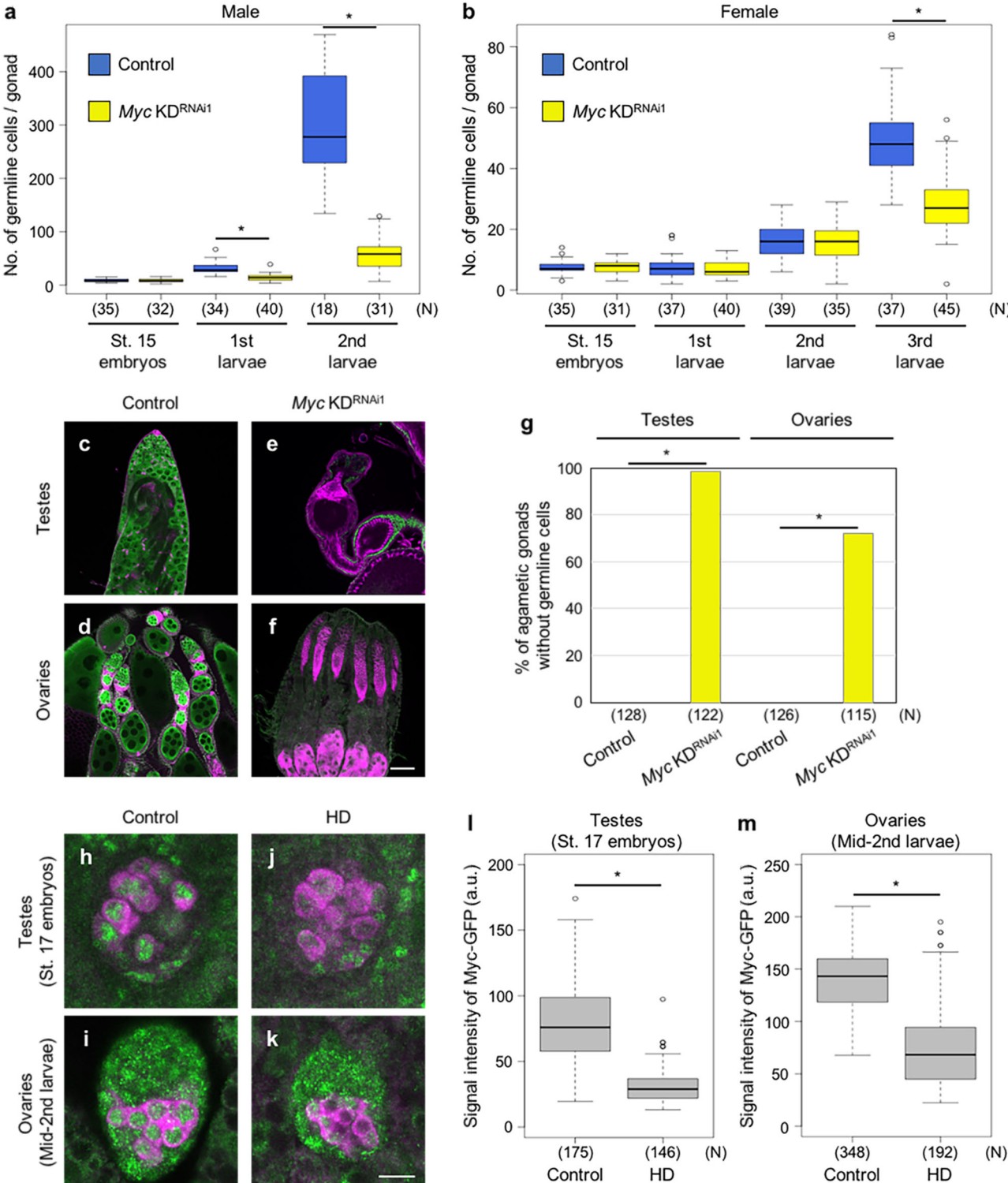

two P-elements into the sn promoter locus[38,39]. When P-element mobilization occurs in the HD germline, the sn[w] allele becomes highly mutable, producing either a wild-type (sn[+]) or extreme singed (sn[e]) phenotype in its offspring[38] (Supplementary Fig. 6). Myc overexpression in the HD male germline caused a significant increase in the percentage of the offspring with the sn[+] and sn[e] phenotypes, relative to offspring from HD males not over-expressing Myc in the germline (Fig. 3a and Supplementary Table 1). Furthermore, Myc overexpression in HD germline significantly decreased the percentage of the offspring that

developed to adulthood, relative to control HD offspring not overexpressing Myc (Fig. 3b, c and Supplementary Table 2). By contrast, under non-HD conditions, neither mutation frequency nor the percentage of offspring developing to adulthood was affected by Myc overexpression (Supplementary Fig. 7 and Supplementary Tables 3 and 4). These observations indicate that inhibition of MDRG by overexpression of Myc in HD germline promotes the production of aberrant gametes that carry highly mutable genes and are incapable of developing properly to adulthood. It is unlikely that Myc overexpression merely represses

**Fig. 2 Phenotype caused by *Myc* knockdown and expression of Myc-GFP in HD germline.** The number of germline cells of male (**a**) and female (**b**) progeny derived from *nos-Gal4/nos-Gal4* females mated with *nos-Gal4/nos-Gal4* (control; blue) and *UASt-Myc^RNAi1^/UASt-Myc^RNAi1^* [*Myc* KD^RNAi1^ (BDSC36123); yellow] males at early embryonic stage 15 and early-first, second, and third instar stages. Gonads were stained for Vasa, and Vasa-positive germline cells in gonads were counted. Each box plot represents median value and first and third quartile values. Error bars represent minimum and maximum values. White circles represent outliers. Significance was calculated by two-sided Student's *t* test; *P < 0.05. *N*: total number of the observed gonads. Similar results were obtained from two biologically independent experiments. Testes (**c**, **e**) and ovaries (**d**, **f**) of control and *Myc* KD^RNAi1^ progeny. Gonads were obtained from adults 3–5 days after eclosion, and stained for Vasa (green), Hts (magenta), and Fasciclin III (magenta). Scale bar: 50 μm. **g** Percentage of agametic gonads without germline cells in control and *Myc* KD^RNAi1^ progeny. Vasa-positive germline cells in gonads were counted. Significance was calculated by two-sided Fisher's exact test, *P < 0.05. *N*: total number of the observed gonads. Similar results were obtained from two biologically independent experiments. Testes at the embryonic stage 17 (**h**, **j**) and ovaries at the mid-second instar stage (**i**, **k**) in progeny derived from *Myc-GFP/Myc-GFP* females mated with *y w* (control; **h**, **i**) and *Harwich* (HD; **j**, **k**) males. Gonads were stained for GFP (green) and Vasa (magenta). Scale bar: 10 μm. Signal intensity of Myc-GFP in germline cell nuclei of control and HD progeny in testes at the embryonic stage 17 (**l**) and ovaries at the mid-second instar stage (**m**). Each box plot represents median value and first and third quartile values. Error bars represent minimum and maximum values. White circles represent outliers. Significance was calculated by two-sided Student's *t* test; *P < 0.05. *N*: total number of the observed gonads. Similar results were obtained from two biologically independent experiments.

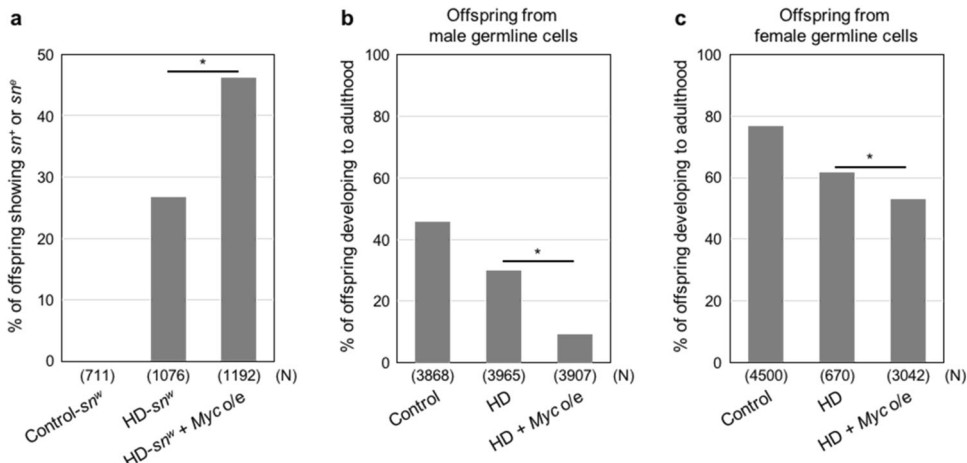

**Fig. 3 *Myc* overexpression in HD germline promotes production of aberrant gametes. a** Mutation frequency in the germline was examined by using the *sn^w^* allele. The mating scheme is shown in Supplementary Table 1 and Supplementary Fig. 6. Control-*sn^w^*, HD-*sn^w^*, and HD-*sn^w^* + *Myc* o/e males were mated with attached-X females. Percentages of male offspring with *sn^e^* and *sn^+^* phenotypes are shown. These phenotypes do not arise unless *sn^w^* is mutated by *P*-element mobilization[38]. Significance was calculated by two-sided Fisher's exact test; *P < 0.05. *N*: total number of the observed offspring. Similar results were obtained from two biologically independent experiments. **b**, **c** Percentage of offspring that developed to adulthood. Offspring were produced from control, HD, and HD + *Myc* o/e males (**b**) and females (**c**) mated with *y w* females and males, respectively. Mating scheme is shown in Supplementary Table 2. Significance was calculated by two-sided Fisher's exact test; *P < 0.05. *N*: total number of the observed offspring. Similar results were obtained from two biologically independent experiments.

mobilization of *P*-elements in the HD germline, because *Myc* overexpression in the HD germline increased mutation frequency or lethality in the offspring (Fig. 3a–c). Therefore, it is reasonable to consider that MDRG eliminates germline cells in which the genetic material has been injured by *P*-element mobilization.

**Roles of apoptosis, *p53*, and *loki* in MDRG.** Although DNA damage associated with HD is thought to cause the germline-loss phenotype, it remains unclear how DNA damage causes germline elimination[17,40]. DNA damage induces cellular responses, including cell-cycle arrest and apoptosis, mediated by the DDR[41,42]. Thus, it is possible that cell-cycle arrest and apoptosis induced by DDR are major causes of MDRG.

To address this hypothesis, we first investigated whether cell-cycle arrest was induced in HD germline at the larval stages, during which the number of germline cells is reduced by HD and *Myc* knockdown. Because cell-cycle arrest is inferred from reduction in the number of mitotic cells, staining for a mitotic marker, phospho-histone H3 (pH3), was performed[18]. The percentages of HD germline cells expressing pH3 were statistically equivalent to those in controls (Supplementary Fig. 8a, b).

Furthermore, *Myc* overexpression did not affect the proportion of HD germline cells that expressed pH3 (Supplementary Fig. 3a, b). Together, these observations suggest that cell-cycle arrest is not a major cause of MDRG at the larval stages. Consistent with this, in males, *Myc* knockdown did not affect the percentage of germline cells with a pH3 signal at the first instar stage (Supplementary Fig. 8c). However, *Myc* knockdown decreased pH3 expression in female germline cells at the third instar. (Supplementary Fig. 8d). These results suggest that *Myc* downregulation did not cause cell-cycle arrest in male germline cells, but in female ones. Therefore, we cannot completely exclude the possibility that cell-cycle arrest is responsible for MDRG.

Second, we found that apoptosis was not required for elimination of germline cells caused by MDRG. Apoptosis is inhibited by overexpression of *Drosophila* inhibitor of apoptosis (*Diap1*)[43]. Hence, in these experiments, we used the *nos-Gal4* driver to induce *UASt-Diap1* in the germline prior to gametogenesis[36,37]. The germline-loss phenotype caused by HD and *Myc* knockdown was not rescued by expression of *Diap1* in either sex (Supplementary Figs. 9 and 10). Therefore, it is unlikely that apoptosis is the cause of MDRG.

Finally, we sought to determine whether *p53* and *loki*/Chk2, which are key regulators that induce cell-cycle arrest and apoptosis in the DDR[44–46], are required for MDRG. To this end, we used *p53*[5A-1–4] and *loki*[P6], viable and fertile loss-of-function mutations. Supplementary Figs. 10 and 11 show that the germline-loss phenotype caused by *Myc* knockdown was not rescued by introduction of *p53*[5A-1–4] and *loki*[P6] in either sex. These observations suggest that *p53* and *loki*/Chk2 are not essential for MDRG, although we cannot rule out the possibility that *Myc* is downstream of *p53* and *loki*/Chk2. Together, these findings suggest that cell-cycle arrest and apoptosis induced by DDR through activity of *loki*/Chk2 and *p53* are not major causes of MDRG.

## Discussion

In this report, we demonstrate that HD causes downregulation of *Myc* expression in the germline, and the *Myc* knockdown results in elimination of germline cells. Conversely, overexpression of *Myc* partially rescues germline elimination in HD larvae and adults. Moreover, the gametes produced from *Myc*-expressing HD germline exhibit a higher mutation frequency and a lower capacity to develop into adulthood than HD germline in the absence of *Myc* overexpression. These observations strongly suggest that MDRG serves to eliminate aberrant germline cells harboring DNA damage due to *P*-element mobilization. However, it is possible that other genes might also be involved in HD-induced germline elimination. Future studies comparing transcriptomes between non-HD and HD germline help us to identify such critical genes.

It remains unclear how MDRG is induced by HD and how MDRG leads to germline cell elimination. Because it is thought that DNA damage associated with HD causes the germline-loss phenotype[17,40], one possible mechanism underlying MDRG is that *Myc* mediates DDR, which induces cell-cycle arrest and apoptosis in the germline[41,42]. In *Drosophila* oogenesis, when female GSCs and their daughter cells are exposed to DNA damage, *loki*/Chk2 is required for cell-cycle arrest of GSCs and the loss of these cells[20,22]. Moreover, in meiotic cells, *p53* acts downstream of *loki*/Chk2 to induce apoptosis in response to DNA damage[18,21]. Thus, germline loss during oogenesis in HD females results from GSC loss and meiotic cell apoptosis, which are caused by DNA damage associated with *P*-element mobilization[22,26]. However, MDRG occurs in HD females at the early third-instar stage (Figs. 1i and 2b, k, m), when GSCs and meiotic cells have not yet been established[47]. Furthermore, mutations in neither *loki*/Chk2 nor *p53* modulate MDRG (Supplementary Figs. 10 and 11). These observations suggest that MDRG is not mediated by DDR. Thus, we propose that germline elimination is regulated in females by at least two different mechanisms, one of which depends on DDR during oogenesis, and the other (MDRG) is independent of DDR at the early third-instar stage. Recent studies reported that malformation of mitochondria is induced in the germline by HD[28], and mitochondrial dysfunction causes downregulation of *Myc* in human cell lines[48]. Hence, future studies focused on the role of mitochondria in MDRG might be an important step toward elucidating how *Myc* mediates elimination of the HD germline.

We found that *Myc* overexpression suppressed the germline-loss phenotype caused by HD, but the suppression was incompletely penetrant, such that approximately half of adult gonads still exhibited the germline-loss phenotype (Fig. 1g). The incomplete penetrance suggests that a pathway active in the HD germline, other than MDRG, is partly responsible for the germline-loss phenotype. Compatible with this, mutation in *loki*/Chk2 partially suppresses the germline-loss phenotype

caused by HD[22]. Therefore, it is reasonable to speculate that at least two germline-elimination mechanisms, MDRG and a Chk2-dependent pathway, are responsible for the HD-induced germline-loss phenotype in females. We propose that MDRG is initiated by *Myc* downregulation in the germline at the larval stages, whereas DDR pathway causes GSC loss and apoptosis in early meiotic cells in response to DNA damage, as reported previously[18]. The reason why overexpression of *Diap1* did not rescue the germline-loss phenotype in HD is that we used UASt to drive expression of *Diap1*, which is activated by *nos-Gal4* in the germline at the embryonic and larval stages (Supplementary Fig. 4), but not during gametogenesis[36,37]. Thus, our *UASt-Diap1* construct was unable to express Diap1 in meiotic cells, where apoptosis was evident[36,37]. We hypothesized that the germline cells with DNA damage are eliminated by a double-check mechanism; at the larval stages by MDRG, and later at gametogenesis by Chk2 activity.

Our observations strongly suggest that elimination of germline cells mediated by *Myc* downregulation maintains the germline integrity. Recent study demonstrates that higher inheritance of germ plasm correlates well with higher survival probability of primordial germ cell (PGC) in embryos[49]. This mechanism ensures selection of the PGCs with higher quantity of germ plasm. However, it is unlikely that this PGC selection depends on MDRG, because MDRG occurs in larvae, but not in embryos (Figs. 1g and 2a, b). We propose that MDRG acts as a quality-control mechanism in the germline, which preserves the genomic integrity of the gametes.

## Methods

**Flies**. *nos-Gal4-VP16* (*nos-Gal4*)[32] was used to induce expression of the following UAS constructs in the germline: *UASt-Myc*[50] (gift from Dr Orian, Dr Eisenman, and Dr Parkhurst); *UASt-Myc*[RNAi1] (36123), *UASt-Myc*[RNAi2] (51454), *UASt-EGFP* (6658), and *UASt-Diap1* (6657) from the Bloomington *Drosophila* Stock Center (BDSC). To examine Myc protein expression, we used the *Myc-GFP* BAC transgenic strain (BDSC, 81274), which underwent recombination to generate a GFP-tagged *Myc* transgene within the genomic DNA surrounding the *Myc* locus[33–35]. To induce HD, males of the P strain *Harwich* (BDSC, 4264) were used. *p53*[5A-1-4] (BDSC, 6815) and *lok*[P6] (gift from Dr Ting Xie), loss-of-function alleles of *p53* and *loki* (Chk2), respectively, were used in the rescue experiment. To examine the mutation frequency in the germline, *singed-weak* (*sn*[w]; 101251) and attached-X chromosome (105722) strains from the *Drosophila* Genomics and Genetic Resources (Kyoto stock center) were used.

Because the temperature sensitivity of HD differs between the sexes[51], male and female progeny derived from HD crosses were cultured at 29 and 25 °C, respectively. *Myc*-knockdown flies were cultured at 30 °C. *loki*[P6] progeny were cultured at 25 °C, because when they were cultured at 30 °C, most of the animals died during larval and pupal development.

**Immunostaining**. Immunofluorescence staining of embryos and gonads was performed as described[52,53]. Embryos at stage 15–17 [10 ± 2 h after egg laying (AEL) (at 29 °C) or 15 ± 3 h AEL (at at 25 °C)] were collected and dechorionated in a sodium hypochlorite solution. Gonads were dissected from early-first [23 ± 2 h AEL (at 29 or 30 °C) or 30 ± 2 h AEL (at 25 °C)], early-second [47 ± 2 h AEL (at 29 or 30 °C) or 54 ± 2 h AEL (at 25 °C)], mid-second [58 ± 2 h AEL (at 25 °C)], and early-third [71 ± 2 h AEL (at 29 or 30 °C) or 78 ± 2 h AEL (at 25 °C)] instar larvae, and adult flies 3–5 days after eclosion. The dechorionated embryos and dissected gonads were fixed in 1:1 heptane:fixative [4% paraformaldehyde in PBS (130 mM NaCl, 7 mM Na$_2$HPO$_4$, and 3 mM NaH$_2$PO$_4$)] for 30 and 15 min, respectively. Vitelline membrane of fixed embryos was removed in 1:1 methanol:heptane by vigorous shaking. The embryos and gonads were washed with PBST (0.1% Triton-X100 in PBS) three times for 15 min each, and then incubated in blocking solution (2% BSA, 0.1% Tween 20, and 0.1% Triton-X100 in PBS) for 1 h. After blocking, the embryos were incubated in the blocking solution containing the following primary antibodies overnight at 4 °C with the indicated dilutions: chick anti-Vasa[54] (1:500), mouse anti-Hts [1:10; Developmental Studies Hybridoma Bank (DSHB), AB_528070], mouse anti-Fasciclin III (1:10; DSHB, AB_528238), rabbit anti-GFP (1:500; Thermo Fisher Scientific, A11122), and rabbit anti-phospho-Histone H3 (Ser10) (1:500; Merck Millipore, 06-570). Then, the embryos and gonads were washed with PBST three times for 15 min each. For detection of primary antibodies, the following secondary antibodies were used: goat anti-chick conjugated with Alexa Fluor 488 (1:500; Thermo Fisher Scientific, A11039) or Alexa Fluor 633 (1:500; Thermo Fisher Scientific, A21103), goat anti-rabbit conjugated with Alexa

Fluor 488 (1:500; Thermo Fisher Scientific, A11034), and goat anti-mouse conjugated with Alexa Fluor 546 (1:500; Thermo Fisher Scientific, A11030). To enhance the Myc-GFP signal, Can Get Signal Immunostain Solution B (Toyobo, NKB-601) was used to dilute anti-GFP antibody.

To examine the signal intensity of Myc-GFP, GFP fluorescence images of the germline cells within a 20-μm range of the embryonic or ovarian upper surface were captured on confocal microscopy SP5 (Leica Microsystems) under the same laser power and gain conditions. Signal intensity of Myc-GFP in the germline nuclei was measured using the FIJI software[55].

**In situ hybridization**. Whole-mount in situ hybridization of embryos was performed as described[52,53]. To synthesize RNA probes for Myc-GFP and Myc mRNA, the EGFP-coding region and *Myc* cDNA were amplified from pEGFP-N1 (Clontech) and an embryonic cDNA library[56], respectively, using the following primer pairs: EGFP-Fw (5′-TATAGGGCGAATTGGGTACCATGGTGAGCAAG GGCGAG-3′)/EGFP-Rv (5′-TCGAGGGGGGGGCCCGGTACCTTACTTGTACA GCTCGTCCATG-3′) and Myc-Fw (5′-TATAGGGCGAATTGGGTACCACCAG ACACACCTCACAGTCTTAC-3′)/Myc-Rv (5′-TCGAGGGGGGGGCCCGGTACC CTGCATACTAAGCTCCTTCTCCTC-3′). Amplified cDNAs were cloned into the *Kpn*I site of pBlueScript II SK+ vector using the In-Fusion HD Cloning Kit (Clontech, 639648). Templates for RNA probes were amplified from these plasmids using the T7 and T3 primers. Digoxigenin (DIG)-labeled RNA probes were synthesized from the fragments using T7 or T3 RNA polymerase (Roche, 10881767001 and 11031163001, respectively).

Embryos at stage 17 [13 ± 1 h AEL (at 29 °C)] were collected and dechorionated in a sodium hypochlorite solution. The dechorionated embryos were fixed and devitellinized as described above. The embryos were then rinsed with ME [50 mM EGTA (pH 8.0) in 90% methanol] and incubated in 7:3, 5:5, and 3:7 ME:fixative for 5 min each. They were then incubated in fixative for 20 min and washed three times with PBSTw (0.1% Tween 20 in PBS). The washed embryos were incubated for 3 min in 50 μg/ml Proteinase K in PBSTw, and the digestion was stopped by incubation for 20 min in fixative, followed by washes three times in PBSTw for 10 min each. The re-fixed embryos were incubated in Pre-HS [50% formamide, 5 × SSC (1 × SSC is 150 mM NaCl and 15 mM sodium-citrate), 100 μg/ml heparin, 100 μg/ml yeast tRNA, 10 mM DTT, and 0.1% Tween 20] for 1 h at 60 °C, and then hybridized with 2 ng/μl of RNA probe in HS (Pre-HS containing 10% dextran sulfate) overnight at 60 °C. After hybridization, the embryos were washed six times at 60 °C for 30 min each in a solution containing 50% formamide, 5 × SSC, and 0.1% Tween 20. Signals were detected using a horseradish peroxidase-conjugated anti-DIG antibody (Roche, 11207733910), and amplified using the TSA Biotin System (PerkinElmer, NEL700001KT) and streptavidin conjugated with Alexa Fluor 546 (Thermo Fisher Scientific, S11225). In situ hybridization combined with immunostaining was performed as described[53]. The embryos were stained with chick anti-Vasa antibody[54] (1:100). Goat anti-chick conjugated with Alexa Fluor 633 (1:500, Thermo Fisher Scientific, A21103) was used as the secondary antibody.

To examine the signal intensities of Myc-GFP and Myc mRNAs, their fluorescence images of the germline cells within a 20-μm range of the embryonic upper surface were captured on an SP5 confocal microscope (Leica Microsystems) under the same laser power and gain conditions. Signal intensities of Myc-GFP and Myc mRNAs in the germline were measured using the FIJI software[46].

**Examination of mutation frequency using the *sn^w* allele**. Male progeny with the *sn^w*, non-*Ser*, and non-*Sb* phenotype, derived from *sn^w*/+; *nos-GAL4/TM3, Ser Sb* females mated with *y w* males (control-*sn^w*), *sn^w*/+; *nos-GAL4/TM3, Ser Sb* females mated with *Harwich* males (HD-*sn^w*), and *sn^w*/+; *nos-GAL4, UASt-Myc/TM3, Ser Sb* females mated with *Harwich* males (HD-*sn^w* + *Myc* o/e), were cultured at 29 °C (Supplementary Table 1) and mated with attached-X (XX/Y) females. The offspring were allowed to develop at 25 °C. Then, adult males exhibiting *singed-weak* (*sn^w*), *singed-extreme* (*sn^e*), and wild-type (*sn^+*) phenotypes were counted. To examine *sn^w* mutation frequency under non-HD conditions, male progeny with the *sn^w*, non-*Ser*, and non-*Sb* phenotype, derived from *sn^w*/+; *nos-GAL4/TM3, Ser Sb* females mated with *y w* males (non-HD_*sn^w*) and *sn^w*/+; *nos-GAL4, UASt-Myc/TM3, Ser Sb* females mated with *y w* males (non-HD_*sn^w* + *Myc* o/e), were cultured at 29 °C (Supplementary Table 3) and mated with attached-X (XX/Y) females. The offspring were allowed to develop at 25 °C, and adult males exhibiting *sn^w*, *sn^e*, and *sn^+* phenotypes were counted.

**Determination of the rate of development to adulthood**. Male and female progeny derived from *nos-Gal4/nos-Gal4* females mated with *y w* males (control) and *nos-Gal4/nos-Gal4* females mated with *Harwich* males (HD) were cultured at 29 and 25 °C, respectively (Supplementary Table 2). Males and females with the non-*Ser* and non-*Sb* phenotype derived from *nos-GAL4, UASt-Myc/TM3, Ser Sb* females mated with *Harwich* males (HD + *Myc* o/e) were cultured at 29 and 25 °C, respectively (Supplementary Table 2). These male and female progeny were mated with *y w* females and males, respectively. Flies were allowed to lay eggs for 16 h on grape juice plates supplemented with yeast paste at 25 °C. Eggs were collected and transferred to vials filled with a standard *Drosophila* medium (100 eggs/vial) and cultured at 25 °C.

The rate at which progeny developed to adulthood under non-HD conditions was examined as follows. Male and female progeny derived from *nos-Gal4/nos-Gal4* females mated with *y w* males (control) were cultured at 29 and 25 °C, respectively (Supplementary Table 4). Males and females with the non-*Ser* and non-*Sb* phenotype derived from *nos-GAL4, UASt-Myc/TM3, Ser Sb* females mated with *y w* males (*Myc* o/e) were cultured at 29 and 25 °C, respectively (Supplementary Table 4). These male and female progeny were mated with *y w* females and males, respectively. Flies were allowed to lay eggs for 16 h on grape juice plates supplemented with yeast paste at 25 °C. Eggs were collected and transferred to vials filled with a standard *Drosophila* medium (100 eggs/vial) and cultured at 25 °C. The frequency of eclosion was calculated.

**Statistics and reproducibility**. Statistical tests conducted in each experiment are indicated in the figure legends. For all experiments, *P* values < 0.05 is considered to be statistically significant. Statistical significance is marked with * (*P* < 0.05) and ns (not significant). All attempts at replication were successful, except for Supplementary Fig. 8c. The percentage of pH3-positive germline cells was slightly downregulated (Experiment 1) and upregulated (Experiment 2) by *Myc* knockdown. The data combining these two results and statistical significance are shown in Supplementary Fig. 8c. The data for *Myc* KD$^{RNAi1}$ + *p53*- in Supplementary Fig. 10 was obtained from a single experiment, due to a failure in maintaining *Myc* KD$^{RNAi1}$ + *p53*- line. Sample sizes were determined based on published studies in this field or our published data.

**Reporting summary**. Further information on research design is available in the Nature Research Reporting Summary linked to this article.

## Data availability
All source data underlying the bar graphs and box plot presented in the figures are reported in Supplementary Data 1. All data and materials produced by this study are available from the corresponding author upon request.

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

## Acknowledgements

We thank Dr Amir Orian, Dr Robert Eisenman, and Dr Susan Parkhurst for the *UASt-Myc* flies and Dr Ting Xie for the *loki*[P6] stock. We also thank the Bloomington *Drosophila* Stock Center and the *Drosophila* Genomics and Genetic Resources for providing us with fly stocks, and the Developmental Studies Hybridoma Bank for antibodies. This work was supported in part by Grants-in-Aid for Scientific Research from the Japan Society for the Promotion of Science (JSPS) (KAKENHI Grant Numbers: 25114002, 24247011, and 18H05552).

## Author contributions

R.O. and S.K. designed the experiments. R.O. performed the experiments. R.O. and S.K. wrote the paper.

## Competing interests

The authors declare no competing interests.
