## [Peer Review File · Communications Biology]

Reviewers' comments:

Reviewer #1 (Remarks to the Author):

Although it is known that P-element mobilization leads to germline loss during hybrid dysgenesis (HD), the underlying molecular mechanism remains unclear. Here the authors study the role of *myc*, a transcriptional factor, in the germ cell loss observed during HD. They found that *myc* was downregulated during HD in germ cells, and that *myc* knock-down in non-dysgenic females phenocopies the dysgenic phenotype. Furthermore, they demonstrate that overexpressing *myc* partially rescued this phenotype during HD, but produced gametes with increased mutation rate, most of which were incapable of developing into adulthood. Finally they show that *Myc* downregulation dependent loss of germline stem cells is independent of cell cycle arrest and apoptosis. This work advances our understanding of the cellular mechanisms that explain P-element hybrid dysgenesis. We feel that these insights are both meaningful and well supported by the presented data. We have no technical concerns about the experimental approaches or results. However, we are uncertain that the work is of interest to a general audience such as the readership of communications biology. Furthermore, the manuscript requires significant revision before it should be published in any journal. The introduction, results and discussion require significant expansion and restructuring to fully motivate the presented research, fully describe the observations, and place the results in the broader context of the field. Additionally there are multiple misinterpretations of the literature on DNA damage response in the female germline which must be addressed.

Comments on writing and structure

Introduction:

The introduction needs to be expanded considerably to describe the previous results motivating this work. Furthermore, the authors need to communicate the significance of their work to a general audience. Specific suggestions follow.

Authors need to describe what previous observation motivated them to examine *Myc*'s function in dysgenic germlines.

Authors should describe the relevant literatures on GSC loss under hybrid dysgenesis, citing, but not limited to the following studies. Some of these works are cited in the intro but their contributions to our understanding of germline loss are minimized or not described (Dorogova et al 2017), other are not mentioned until the discussion (Tasnim and Kelleher 2017) and other are not cited at all (Schaefer et al 1979, Kelleher et al 2018).

Dorogova N V., Bolobolova EU, Zakharenko LP. 2017. Cellular aspects of gonadal atrophy in *Drosophila* P-M hybrid dysgenesis. *Dev Biol* 424: 105–112.

<http://www.ncbi.nlm.nih.gov/pubmed/28283407> (Accessed June 19, 2017).

Tasnim S, Kelleher ES. 2017. p53 is required for female germline stem cell maintenance in P- element hybrid dysgenesis. *Dev Biol*. <http://linkinghub.elsevier.com/retrieve/pii/S0012160617307327> (Accessed January 3, 2018).

Kelleher ES, Jaweria J, Akoma U, Ortega L, Tang W. 2018. QTL mapping of natural variation reveals that the developmental regulator *bruno* reduces tolerance to P-element transposition in the *Drosophila* female germline. *H. Malik. PLOS Biol* 16: e2006040.

<http://dx.plos.org/10.1371/journal.pbio.2006040> (Accessed November 1, 2018).

Schaefer RE, Kidwell MG, Fausto-Sterling A. 1979. Hybrid Dysgenesis in *Drosophila melanogaster*: Morphological and Cytological Studies of Ovarian Dysgenesis. *Genetics* 92: 1141–52.

<http://www.pubmedcentral.nih.gov/articlerender.fcgi?artid=1214061&tool=pmcentrez&rendertype=abstract> (Accessed June 4, 2012).

The intro should also provide more information on what is known about DNA response in the female germline at different stages of gametogenesis, with a particular focus on the earliest pre-meiotic stages where the P-element is mobile. The following works which the authors cite in the discussion, should be cited discussed in the intro in the context of DNA damage response as a determine of GSC maintenance.

Shim HJ, Lee E-M, Nguyen LD, Shim J, Song Y-H. 2014. High-dose irradiation induces cell cycle arrest, apoptosis, and developmental defects during *Drosophila* oogenesis. *PLoS One* 9: e89009. <http://www.ncbi.nlm.nih.gov/pubmed/24551207> (Accessed June 21, 2017).

Ma X, Han Y, Song X, Do T, Yang Z, Ni J, Xie T. 2016. DNA damage-induced Lok/CHK2 activation compromises germline stem cell self-renewal and lineage differentiation. *Development* 143: 4312–4323. <http://www.ncbi.nlm.nih.gov/pubmed/27729408> (Accessed February 26, 2018).

Wylie A, Lu W-J, D?Brot A, Buszczak M, Abrams JM. 2014. p53 activity is selectively licensed in the *Drosophila* stem cell compartment. *Elife* 3: e01530. <http://www.ncbi.nlm.nih.gov/pubmed/24618896> (Accessed June 21, 2017).

Results:

The discussion cites many additional results that are presented in supplementary figures, including staining for meiotic markers (Figure S3) and effects of suppressing apoptosis (Figures S4 and S5). These data should be presented in the results section since they test specific hypotheses about Myc's potential role in suppressing cell-cycle arrest and apoptosis. I assume the authors left these results to these discussion because they are "negative results". However they are are very important for understanding the biology of Myc downregulation in damaged germline stem cells.

The authors have used same number of symbols (*) for indicating different values of statistical significance at different places. For example, in figures (1-3), * denotes P-value < 0.05 while in supplemental figures (2-3), * denotes P-value > 0.05. It is recommended to use the standard connotation for indicating statistical significance or be consistent in the use of one (using * for P-value < 0.05 throughout the paper) to avoid confusion.

Review of previous literature and data interpretation:

From the beginning of the manuscript the authors need to delineate differences in the effects of P-element activity and different stages of organismal and oogenetic stages. For example, embryonic and larval phenotypes observed under restrictive conditions (29C) do not necessarily have the same mechanistic basis as those observed in adults reared under permissive conditions (25C), yet these phenotypes are presented interchangeably in the manuscript. Dorogova et al (2017) nicely describes distinctions between the two. Furthermore, even within the adult ovary, the effect of P-element activity differs between germline stem cells (where it drives their loss), meiotic cells (where it drives apoptosis and cell cycle arrest). The adult data in this manuscript addresses the earliest GSC effects of P-elements, and that should be made clear to the readers.

Related to above, the description of apoptosis in the adult female germline (currently in the discussion) is very misleading. In adult ovaries, female GSCs do not undergo apoptosis (Hassell 2013, Ma et al 2016). Thus it was known prior to this manuscript that the loss of GSCs under dysgenic conditions is apoptosis independent (Tasnim and Kelleher 2018). As a point of clarification, previous studies have shown that meiotic cysts, which are apoptosis competent, do undergo cell-death in the ovaries of dysgenic females (Dorogova 2017). Critically, this also means that the observation that suppressing apoptosis in the germline of embryos and larvae does not suppress GSC loss (Figure S4)

is in no way inconsistent with previous observations, which differs from what the authors assert in the discussion (Page 12, paragraph 2).

The discussion of p53's function in damaged GSCs is also quite misleading. The specific function of p53 in adult female GSCs unclear, but it appears to act antagonistically to GSC loss in the event of DNA damage, and does not induce apoptosis (Wylie et al 2014, Ma et al 2016). Thus, the observation that p53 alleles don't rescue hybrid dysgenesis is not unexpected, and in fact has been demonstrated previously (Tasnim and Kelleher 2018). In addition to clearly presenting these previous observations, the authors also should not argue that their observation that p53 mutations do not suppress hybrid dysgenesis is novel (Figure S6) as they currently do in the discussion pages (pages 12-13).

The authors hypothesize that a Myc-dependent pathway determines germline loss in embryos and larvae, while their fate in adults is determined by Chk-2 and p53 (discussion, page 14). In addition to the incorrect description of p53's function in adult GSCs, it is not entirely clear to us that their experiments rule out a role for Chk2 and p53 for influencing GSC loss in the larval gonad. Because p53 mutant alleles enhance GSC loss in adults it is not informative to look at the effects of Myc KD in p53 mutants. The appropriate experiment would be to overexpress p53 concurrently with Myc KD, or perhaps to do Myc KD in Chk2 mutants (which legitimately suppress dysgenesis).

Reviewer #2 (Remarks to the Author):

The authors explore the causes for the reduced germline viability in hybrid dysgenesis crosses. They present evidence to suggest that the P-element mobilization taking place in these crosses leads to reduced levels of Myc, which in turn may trigger the demise of these affected germline cells. Unfortunately, neither the effect of P-elements on Myc protein levels, nor the rescue of the reduced germline viability are fully convincing.

First, to monitor Myc levels the authors exclusively rely on a Myc:GFP transgene. This construct has been described in one publication as being able to rescue the lethality of a Myc null allele, i.e. it produces a functional Myc protein. However, it is not known whether Myc:GFP also rescues the sterility of Myc mutations – hence it has not been demonstrated that the expression pattern of Myc:GFP in the germline reflects the expression pattern of endogenous Myc. Furthermore, the fact that a Myc transgene rescues the lethality of a Myc null mutation does not imply that this transgene faithfully reproduces the expression pattern of endogenous Myc - ubiquitously expressed Myc transgenes (e.g. driven by the tubulin promoter) have also been published to rescue a Myc null allele.

Furthermore, it is surprising that a ca. 50% decrease in Myc levels (as described for Myc:GFP in HD ovaries in Fig. 2m) should be sufficient for the near total loss of germ line in this setting. After all, Myc heterozygotes (null/wt) are fertile, and hypomorphic Myc alleles which reduce Myc (transcript) levels to a similar level (e.g. dm[1]) allows the normal development of early adult ovaries (these ovaries degenerate only later on).

In light of these concerns, the authors need to determine Myc levels in the various settings by at least one other means, i.e. antibody staining, ISH, or qPCR.

Second, it is unclear to which extent the UAST-system allows GAL4-driven overexpression in the germ line. The publications cited by the authors (Rorth 1998, DeLuca & Spradling 2018) have emphasized that such elements generally do not work in the germ line. In light of this, the authors need to make sure that "Myc overexpression" really results in increased Myc levels in the germ line.

Additional minor issues:

1. the authors state that Myc-overexpression does not decrease "mutation rates" or "lethality" in HD offspring. They need to present a figure that explicitly shows this.
2. the percentage of "normally developed ovaries" seems to be 0 in Fig. S5, whereas in Fig. 2G only

ca. 70% of Myc-KD ovaries are counted as "agametic". Where does this discrepancy come from?
3. 5 genotypes in Fig. S5 do not have any visible bars for "% normally developed gonads", but high significance levels are indicated for several comparisons between such invisible bars. The authors need to add a figure at a different scale that reveals these invisible bars.

Reviewer #3 (Remarks to the Author):

Ota and Kobayashi argue that Myc mediates germ cell elimination during P-M hybrid dysgenesis (HD) in *Drosophila*. This conclusion is based on several results.

1. Germline knockdown of Myc results in loss of germ cells, as does P-M hybrid dysgenesis. Both processes appear to arrest gametogenesis early, but further degree of phenotypic similarity is not presented or discussed in detail.
2. Myc expression level is lower in HD situations relative to control, based on signal of a Myc-GFP expressed under normal Myc regulation. This would only be meaningful if only cells at the same gametogenic stages are compared, but whether this was done is unclear from the text.
3. Overexpression of Myc in the germline rescues the loss of germ cells.
4. Germ cells resulting from this rescue show a high rate of mutations resulting from P-element mobilization.

My opinion of this paper is mixed. The question under study is interesting. A mechanism for surveillance would be important. The quality of the data is OK. But there are concerns about interpretation and design. The authors' data are consistent with their model, but I do not believe that they have proven it to the exclusion of others.

The most compelling result favoring their model is that Myc overexpression rescues germ cell number in the germline. It is less convincing that Myc is part of the normal surveillance mechanism that eliminates germline cells with genetic damage, for the following reasons:

1. Whether the loss of germ cells in HD is caused by Myc down-regulation is uncertain. Only one RNAi line is used for the knockdown experiments, leaving concern about off-targets. Also, the only molecule that the authors looked at in HD germ cells is Myc. It is possible that some other critical molecule also is mis-regulated in those cells. Finally, as noted above details of phenotypic similarity are not fully clear.
2. Ectopic expression of Myc may bypass normal survival pathways. It does not prove that the reason that the cells normally die in HD is due to Myc decrease.
3. The Discussion presents and discusses supplementary data that the authors believe argue against the Myc mechanism being through apoptosis, that a separate Chk2 pathway synergizes with downregulation of Myc as a separate germline elimination mechanism. These experiments are not presented in the depth needed to evaluate them, and are not sufficient to be certain of the mechanistic interpretation.
4. Line 111: does overexpression of Myc have any effect in non-HD conditions? Do more cells enter gametogenesis?
5. Line 127ff and Fig. 3 what is seen in wild type ovaries with over-expression of Myc? Hopefully no increase in mutation rate.

Other items:

1. Myc effects are seen in both sexes but HD only occurs in the female germline.
2. Starting in line 124, the authors use the abbreviation MDRG, but it is not defined.
3. Statements like "(we) identified the genes involved in the elimination of germ cells" (line 42-43) are too strong.

4. Line 125 and elsewhere, probably better to say "mutation frequency" than "mutation rate"
5. Line 138 and elsewhere: gametes are not "mutable", genes are.
6. Data are needed for the statement in lines 140-141.
7. Lines 252-256, I would expect sons of these crosses to also show mutant phenotypes in sn, but this is not reported

Our point-by-point-responses to the reviewers' comments are provided below. To avoid ambiguity, a single numbering system is used for all reviewer comments: Reviewer 1, Comments #1–8; Reviewer 2, Comments #9–13; Reviewer 3, Comments #14–24.

Reviewer 1

Comment #1

Authors need to describe what previous observation motivated them to examine Myc's function in dysgenic germlines.

According to the reviewer's suggestion, we added the following sentence on page 5, line 65; “Over the course of our experiments to screen transcription factors for germline development and its proliferation, we came across the phenotype caused by knockdown of Myc, a basic helix-loop-helix transcription factor^{29–31}, that was similar to the germline-loss phenotype observed in HD progeny.”

Comment #2

Authors should describe the relevant literatures on GSC loss under hybrid dysgenesis, citing, but not limited to the following studies. Some of these works are cited in the intro but their contributions to our understanding of germline loss are minimized or not described (Dorogova et al 2017), other are not mentioned until the discussion (Tasnim and Kelleher 2017) and other are not cited at all (Schaefer et al 1979, Kelleher et al 2018).

The intro should also provide more information on what is known about DNA response in the female germline at different stages of gametogenesis, with a particular focus on the earliest pre-meiotic stages where the P-element is mobile. The following works which the authors cite in the discussion, should be cited discussed in the intro in the context of DNA damage response as a determine of GSC maintenance.

According to the reviewer's suggestions, we added the following statement on page 4, line 50; “This germline-loss phenotype is thought to be caused by DNA damage associated with P-element mobilization¹⁷. In ovarian germline cells, DNA damage

induces cell-cycle arrest and apoptosis via the functions of two DNA damage response (DDR) genes, *loki* (also known as *mnk*) encoding Checkpoint kinase 2 (Chk2) and *p53*^{18–21}. In early oogenesis, *loki*/Chk2 is required for cell-cycle arrest of the germline stem cells (GSCs) and their daughters in response to DNA damage^{18,20}. By contrast, *p53* is required for cell-cycle re-entry of GSCs and their maintenance^{19,20}. Later, in meiotic cells, *p53* acts as a downstream effector of *loki*/Chk2 to induce apoptosis^{18,21}. Consistent with their roles in the DNA damage response, *loki*/Chk2 and *p53* also act as modifiers of HD-caused germline loss during oogenesis²². Although there is no known link between DNA damage response and *Bruno*, a regulator of oogenesis^{23–25}, *Bruno* activity is also required for the GSC loss phenotype in adult HD females²⁶. By contrast, at the pre-adult stage, the roles of *loki*/Chk2 and *p53* in the germline-loss phenotype caused by HD remain elusive; nevertheless, germline loss is observed from embryonic stage 16 onward^{16,27,28}. Thus, the mechanisms underlying germline elimination in HD are not fully understood.”

Comment #3

The discussion cites many additional results that are presented in supplementary figures, including staining for meiotic markers (Figure S3) and effects of suppressing apoptosis (Figures S4 and S5). These data should be presented in the results section since they test specific hypotheses about Myc’s potential role in suppressing cell-cycle arrest and apoptosis.

According to the reviewer’s suggestion, we moved the data from the Discussion to the Results section on page 12, line 173.

Comment #4

*It is recommended to use the standard connotation for indicating statistical significance or be consistent in the use of one (using * for P -value < 0.05 throughout the paper) to avoid confusion.*

According to the reviewer’s suggestion, we unified indications of statistical significance in this manuscript: “*” for $P < 0.05$ and “ns” for not significant ($P \geq 0.05$).

Comment #5-1

From the beginning of the manuscript the authors need to delineate differences in the effects of P-element activity and different stages of organismal and oogenetic stages. For example, embryonic and larval phenotypes observed under restrictive conditions (29C) do not necessarily have the same mechanistic basis as those observed in adults reared under permissive conditions (25C), yet these phenotypes are presented interchangeably in the manuscript. Dorogova et al (2017) nicely describes distinctions between the two.

According to the reviewer's suggestion, in the revised manuscript, we described differences in the effects of HD or *Myc* function during germline development. For example, please see below (#5-2)

Comment #5-2

Furthermore, even within the adult ovary, the effect of P-element activity differs between germline stem cells (where it drives their loss), meiotic cells (where it drives apoptosis and cell cycle arrest). The adult data in this manuscript addresses the earliest GSC effects of P-elements, and that should be made clear to the readers.

We agree. Accordingly, we added the following sentences on page 15, line 219: "In *Drosophila* oogenesis, when female GSCs and their daughter cells are exposed to DNA damage, *loki/Chk2* is required for cell-cycle arrest of GSCs and the loss of these cells^{20,22}. Moreover, in meiotic cells, *p53* acts downstream of *loki* to induce apoptosis in response to DNA damage^{18,21}. Thus, germline loss during oogenesis in HD females results from GSC loss and meiotic cell apoptosis, which are caused by DNA damage associated with P-element mobilization^{22,26}. However, MDRG occurs in HD females at the early third-instar stage (Fig. 1i and Fig. 2b, k, and m), when GSCs and meiotic cells have not yet been established⁴⁷. Furthermore, mutations in neither *loki/Chk2* nor *p53* modulate MDRG (Fig. S10 and S11). These observations suggest that MDRG is not mediated by DDR. Thus, we propose that germline elimination is regulated in females by at least two

different mechanisms, one of which depends on DDR during oogenesis, and the other (MDRG) is independent of DDR at the early third-instar stage.”

Comment #6-1

Related to above, the description of apoptosis in the adult female germline (currently in the discussion) is very misleading. In adult ovaries, female GSCs do not undergo apoptosis (Hassell 2013, Ma et al 2016). Thus, it was known prior to this manuscript that the loss of GSCs under dysgenic conditions is apoptosis independent (Tasnim and Kelleher 2018). As a point of clarification, previous studies have shown that meiotic cysts, which are apoptosis competent, do undergo cell-death in the ovaries of dysgenic females (Dorogova 2017).

We agree that our original description of apoptosis in the adult female germline was misleading. We changed our statements in the discussion as follows.

Page 15, line 221

“loki and p53 are required for cell-cycle arrest and GSC maintenance, respectively.”

↓

“loki/Chk2 is required for cell-cycle arrest of GSCs and the loss of these cells^{20,22}.”

Page 17, line 244

“MDRG is initiated by *Myc* downregulation in the germline at the larval stages, and is independent of apoptosis and p53 activity. By contrast, the other pathway, which depends on p53 activity, causes apoptosis in early gametogenesis in response to DNA damage, as reported previously.”

↓

“MDRG is initiated by *Myc* downregulation in the germline at the larval stages, and is independent of apoptosis and loki/Chk2 and p53 activity, whereas the other pathway, which depends on Chk2 and p53 activity, causes GSC-loss and apoptosis in early meiotic cells in response to DNA damage, as reported previously¹⁸.”

Page 17, line 253

“at the larval stages by MDRG, and later at gametogenesis by apoptosis.”

↓

“at the larval stages by MDRG, and later at gametogenesis by Chk2 activity.”

Comment #6-2

Critically, this also means that the observation that suppressing apoptosis in the germline of embryos and larvae does not suppress GSC loss (Figure S4) is in no way inconsistent with previous observations, which differs from what the authors assert in the discussion (Page 12, paragraph 2).

As mentioned on page 16, line 246, “The reason why overexpression of *Diap1* did not rescue the germline-loss phenotype in HD is that we used UAS to drive expression of *Diap1*, which is activated by *nos-Gal4* in the germline at the embryonic and larval stages (Fig. S4), but not during gametogenesis^{36,37}. Thus, our *UAS-Diap1* construct was unable to express *Diap1* in meiotic cells, where apoptosis was evident^{36,37}.”

Comment #7-1

*The discussion of *p53*'s function in damaged GSCs is also quite misleading. The specific function of *p53* in adult female GSCs unclear, but it appears to act antagonistically to GSC loss in the event of DNA damage, and does not induce apoptosis (Wylie et al 2014, Ma et al 2016). Thus, the observation that *p53* alleles don't rescue hybrid dysgenesis is not unexpected, and in fact has been demonstrated previously (Tasnim and Kelleher 2018).*

We agree. Accordingly, we rewrote our statements regarding *p53*, mentioned in Comments #6-2 and #7-1, in the Introduction and Discussion sections. However, we did not test whether the *p53* mutation was unable to rescue HD (please see below).

Comment #7-2

In addition to clearly presenting these previous observations, the authors also should not argue that their observation that p53 mutations do not suppress hybrid dysgenesis is novel (Figure S6) as they currently do in the discussion pages (pages 12-13).

This comment may be based on misunderstanding. We only showed that *p53* mutation did not rescue germline-loss phenotype caused by MDRG; we have not examined the role of *p53* in HD. Furthermore, there is no Fig. S6 in the original version of the manuscript.

Comment #8

The authors hypothesize that a Myc-dependent pathway determines germline loss in embryos and larvae, while their fate in adults is determined by Chk-2 and p53 (discussion, page 14). In addition to the incorrect description of p53's function in adult GSCs, it is not entirely clear to us that their experiments rule out a role for Chk2 and p53 for influencing GSC loss in the larval gonad. Because p53 mutant alleles enhance GSC loss in adults it is not informative to look at the effects of Myc KD in p53 mutants. The appropriate experiment would be to overexpress p53 concurrently with Myc KD, or perhaps to do Myc KD in Chk2 mutants (which legitimately suppress dysgenesis).

According to the reviewer's suggestion, we performed the experiment to determine whether *loki/Chk2* mutation affects germline-loss phenotype caused by *Myc* knockdown. The results were added as Fig. S11. Our data show that *loki/Chk2* mutation did not rescue the germline-loss phenotype caused by *Myc* knockdown. Thus, MDRG occurs independently of Chk2 and p53 activity. These data were cited on page 13, line 200 and on page 16, line 228.

Reviewer #2

Comment #9

the authors need to determine Myc levels in the various settings by at least one other means, i.e. antibody staining, ISH, or qPCR.

We agree. We have been trying to obtain an antibody against Myc protein. We raised a new anti-Myc antibody and also purchased an antibody (P4C4-B10) from the Developmental Studies Hybridoma Bank, but they did not work well for our immunostaining of whole embryos. Next, we tried to detect endogenous *Myc* mRNA, as well as *Myc-GFP* mRNA, in the germline. We found that both mRNAs expressed in male germline cells of stage 17 non-HD embryos, and that their levels are reduced in HD embryos (Fig. S3a–e and g). This strongly supports our conclusion that HD reduces Myc expression in germline cells. However, we could not detect these mRNA signals in female germline cells of mid-second instar larvae, presumably due to low permeability of the *Myc* probe. Although we planned to quantify endogenous *Myc* mRNA using qPCR, we were unable to isolate female or male germline cells from HD embryos. This is because it is extremely difficult to introduce a fluorescent marker (that can distinguish male vs. female germline) into the X chromosome of *Harwich* line. To explain, we added the following statement regarding *in situ* hybridization of *Myc* mRNA on page 8, line 116: “Furthermore, the levels of *Myc-GFP* and endogenous *Myc* mRNAs were significantly reduced in the germline cells of HD males at embryonic stage 17 (Fig. S3a–e and g), although we could not detect these mRNAs in the female germline cells at the mid-second instar, presumably because it was difficult for the probes to permeate the gonads. These results suggest that *Myc* expression was reduced in germline cells of HD progeny before germline number severely decreased.”

Comment #10

it is unclear to which extent the UAST-system allows GAL4-driven overexpression in the germ line. The publications cited by the authors (Rorth 1998, DeLuca & Spradling 2018) have emphasized that such elements generally do not work in the germ line. In light of

this, the authors need to make sure that “Myc overexpression” really results in increased Myc levels in the germ line.

Although it has been reported that UAS_T does not work in the germline, we know that it is active in the germline of embryos and larvae. Accordingly, we added the following sentences (page 9, line 128) and data (Fig. S3f and g, and Fig. S4) to the revised manuscript.

“Although UAS_T is not active in the germline during oogenesis^{36,37}, UAS_T-EGFP was activated by nos-Gal4 to produce EGFP protein in the germline of testes at the embryonic stage 17, and in ovaries at the mid-second instar, when Myc expression was reduced by HD (Fig. S4). Using UAS_T-Myc and nos-Gal4, we overexpressed Myc in the germline cells (Fig. S3f and g).”

Comment #11

the authors state that Myc-overexpression does not decrease “mutation rates” or “lethality” in HD offspring. They need to present a figure that explicitly shows this.

To avoid confusion, we rewrote the sentence as follows (page 11, line 169) and added a figure citation: “Myc overexpression in the HD germline increased mutation frequency or lethality in the offspring (Fig. 3a–c).”

Comment #12

the percentage of “normally developed ovaries” seems to be 0 in Fig. S5, whereas in Fig. 2G only ca. 70% of Myc-KD ovaries are counted as “agametic”. Where does this discrepancy come from?

In Fig. S10 (Fig. S5 of the original manuscript), morphologies of the gonads were observed. Ovaries with more than three mature eggs were considered “normal”. By contrast, in Fig. 2g, the ovaries were immunostained, and those without Vasa-positive germline cells were considered “agametic”. To avoid confusion, we added the following sentence to the legend of Fig. S10. “Gonads were obtained from adults 3–5 days after

eclosion, and their morphologies were observed. Ovaries with more than three mature eggs and testes > 1 mm in length were considered normally developed gonads.”

Comment #13

5 genotypes in Fig. S5 do not have any visible bars for “% normally developed gonads”, but high significance levels are indicated for several comparisons between such invisible bars. The authors need to add a figure at a different scale that reveals these invisible bars.

We used * for $P \geq 0.05$ in Fig. S5 of the original version of the manuscript. To avoid misleading, we used ns (not significant) for $P \geq 0.05$ in Fig. S10 of the revised manuscript.

Reviewer 3

Comment #14-1

Whether the loss of germ cells in HD is caused by Myc down-regulation is uncertain. Only one RNAi line is used for the knockdown experiments, leaving concern about off-targets.

We agree. Accordingly, we added Fig. S1 to show the number of germline cells during embryonic and larval development in *Myc* KD using another *Myc* RNAi line. The statement “Similar results were obtained using another shRNA targeting a distinct region of the *Myc* mRNA (Fig. S1a and b).” was added on page 7, line 101.

Comment #14-2

The only molecule that the authors looked at in HD germ cells is Myc. It is possible that some other critical molecule also is mis-regulated in those cells.

In this study, we identified only one gene, *Myc*, as a mediator of HD-induced germline elimination. However, as the reviewer suggest, other genes may also be mis-regulated by HD. Future studies should seek to identify such genes, by comparing transcriptomes between non-HD and HD germline.

Comment #14-3

Details of phenotypic similarity are not fully clear.

We agree. Accordingly, we changed our statements as follows.

Page 7, line 95

“*Myc* KD caused a phenotype very similar to that observed in HD progeny.”

↓

“*Myc* KD caused a reduction in the number of germline cells similar to that observed in HD progeny.”

Comment #15

Ectopic expression of Myc may bypass normal survival pathways. It does not prove that the reason that the cells normally die in HD is due to Myc decrease.

We believe that Myc is a *bona fide* mediator for HD-induced elimination of germline cells at the larval stages because we observed downregulation of Myc in HD germline cells (Fig. 2h–m and Fig. S3), Myc downregulation resulted in germline elimination (Fig. 2a–g and Fig. S1a and b), and *Myc* overexpression rescued the reduction of the number of germline cells in HD (Fig. 1).

Comment #16

The Discussion presents and discusses supplementary data that the authors believe argue against the Myc mechanism being through apoptosis, that a separate Chk2 pathway synergizes with downregulation of Myc as a separate germline elimination mechanism. These experiments are not presented in the depth needed to evaluate them, and are not sufficient to be certain of the mechanistic interpretation.

We agree. We extensively investigated the role of *loki/Chk2* in MDRG, and ultimately found that *loki/Chk2* mutation was unable to rescue the germline-loss phenotype caused by *Myc* knockdown. These data were added as Fig. S11.

Because MDRG occurs independently of *Diap1*, *loki/Chk2*, and *p53*, we added the following sentence on page 17, line 244: “We propose that MDRG is initiated by *Myc* downregulation in the germline at the larval stages, and is independent of apoptosis and *loki/Chk2* and *p53* activity, whereas the other pathway, which depends on *Chk2* and *p53* activity, causes GSC-loss and apoptosis in early meiotic cells in response to DNA damage, as reported previously¹⁸.”

Comment #17

does overexpression of Myc have any effect in non-HD conditions? Do more cells enter gametogenesis?

what is seen in wild type ovaries with over-expression of Myc? Hopefully no increase in mutation rate.

Over the course of the revision process, we found that “By contrast, under non-HD conditions, neither mutation frequency nor the percentage of offspring developing to adulthood was affected by *Myc* overexpression (Fig. S7, and Table S3 and S4).” This statement was added on page 11, line 162. Furthermore, the number of eggs produced by females appeared to be unaffected by *Myc* overexpression under non-HD conditions.

Comment #18

Myc effects are seen in both sexes but HD only occurs in the female germline.

Our observations (Fig. 1c, d, and g) clearly show that HD occurs in both sexes. This has been previously reported (Engels and Preston, *PNAS*, 1978; Kidwell and Novy, *Genetics*, 1979).

Comment #19

Starting in line 124, the authors use the abbreviation MDRG, but it is not defined.

MDRG is defined on page 10, line 147. For clarification, bold underlines were added as follows: “~~*Myc*~~-downregulation-~~dependent~~ reduction in the number of germline cells (MDRG)”

Comment #20

Statements like “(we) identified the genes involved in the elimination of germ cells” (line 42-43) are too strong.

According to the reviewer's suggestion, we changed the sentence on page 3, line 41 as follows:

“we introduced genetic damage into the germline by mobilizing P-elements and identified the genes involved in elimination of germline cells.”

↓

“we induced the elimination of germline cells by mobilizing P-elements and identified genes rescuing the elimination.”

Comment #21

Line 125 and elsewhere, probably better to say “mutation frequency” than “mutation rate”

According to the reviewer's suggestion, we used “mutation frequency” instead of “mutation rate(s)” throughout the manuscript.

Comment #22

Line 138 and elsewhere: gametes are not “mutable”, genes are.

According to the reviewer's suggestion, we rewrote the sentence in page 11, line 166 as follows: “gametes that carry highly mutable genes and are incapable of developing properly to adulthood.”

Comment #23

Data are needed for the statement in lines 140–141.

According to this suggestion, we cited data in this statement as follows (page 11, line 169): “Myc overexpression in the HD germline increased mutation frequency or lethality in the offspring (Fig. 3a–c)”

Comment #24

Lines 252–256, I would expect sons of these crosses to also show mutant phenotypes in sn, but this is not reported.

According to the reviewer's suggestion, we added Fig. S6 showing images of *sn* phenotypes and the mating scheme.

REVIEWERS' COMMENTS:

Reviewer #1 (Remarks to the Author):

This is a revision of a previously submitted manuscript, which we reviewed last year. The paper combines genetic and cytological approaches to study the role of Myc repression in the elimination of germline cells after DNA damage. The authors demonstrate that Myc downregulation promotes GSC loss, and that Myc is down-regulated by P-element hybrid dysgenesis. They further demonstrate that Myc overexpression rescues GSC maintenance in dysgenic gonads, but that the resulting gametes exhibit increased transposon excisions and reduced viability. The authors suggest that Myc downregulation initiates the elimination of damaged GSCs, thereby ensuring production of the highest quality gametes. They further suggest that Myc-downregulation dependent elimination occurs independently of the DNA damage response.

This paper is extensively revised and improved from the previous version, and I'm grateful that the authors carefully considered the suggestions that we made. Nevertheless, we do have one major concern with the way the data are interpreted, that we think the authors should address. We also have some more minor suggestions, including literature they may want to cite to most effectively convey the significance of their work.

Interpretation concerns:

The authors suggest that because Chk2 mutations do not rescue GSC loss resulting from Myc KD, then Chk2 and Myc must act in independent pathways. I agree that the Diap1 experiments suggest that MDRG is apoptosis independent. I also agree that the MyKD in p53/Chk2 mutant results suggest that p53 and Chk2 are not downstream targets of Myc. However, these experiments do not rule out that Myc repression results from Chk2 activation. If Myc is downregulated by Chk2 activation, then Chk2 mutations would have no effect on the phenotype of Myc KD. I suggest the authors revise their discussion to allow for the possibility, and remove any claims that they have conclusively demonstrated that Myc is in a separate pathway from Chk2 and p53.

Minor concerns/suggestions:

For pH3 experiments it would help if authors clearly stated that arrest is inferred from an absence or reduction of pH3 positive cells.

Similarly in discussion of pH3 results, authors might want to consider that male and female germline are different. Myc downregulation could lead to arrest in female germlines but not male.

In the discussion the authors may wish to compare their observations with myc to the role of wunen2 in determining pole cell elimination in the early embryos (Slaidina and Lehmann 2017). The idea is similar, that wunen2 may indicate pole cell quality to ensure the fittest germline in the developing organism, although in the wunen2 case, germ cell elimination appears to rely on p53-dependent apoptosis.

Reviewer #2 (Remarks to the Author):

The authors have addressed my concerns as well as possible. It would have been important to document endogenous Myc levels under HD conditions, but the technical difficulties in doing so should not prevent this interesting paper from being published.

Our point-by-point-responses to the reviewer's comments are provided below.

Reviewer 1

Comment #1

I also agree that the MycKD in p53/Chk2 mutant results suggest that p53 and Chk2 are not downstream targets of Myc. However, these experiments do not rule out that Myc repression results from Chk2 activation. If Myc is downregulated by Chk2 activation, then Chk2 mutations would have no effect on the phenotype of Myc KD. I suggest the authors revise their discussion to allow for the possibility, and remove any claims that they have conclusively demonstrated that Myc is in a separate pathway from Chk2 and p53.

According to the reviewer's suggestion, we changed the statements as follows.

Page 14, line 213:

“suggesting that p53 and loki /Chk2 are not essential for MDRG.”

↓

“These observations suggest that p53 and loki/Chk2 are not essential for MDRG, although we cannot rule out the possibility that Myc is downstream of p53 and loki/Chk2.”

Page 17, line 257:

“Therefore, it is reasonable to conclude that at least two germline-elimination mechanisms,”

↓

“Therefore, it is reasonable to speculate that at least two germline-elimination mechanisms,”

Page 17, line 259:

“We propose that MDRG is initiated by Myc downregulation in the germline at the larval stages, and is independent of apoptosis and loki/Chk2 and p53 activity, whereas

the other pathway, which depends on Chk2 and p53 activity, causes GSC-loss and apoptosis in early meiotic cells in response to DNA damage, as reported previously¹⁸.”

↓

“We propose that MDRG is initiated by *Myc* downregulation in the germline at the larval stages, whereas DDR pathway causes GSC-loss and apoptosis in early meiotic cells in response to DNA damage, as reported previously¹⁸.”

Comment #2

For pH3 experiments it would help if authors clearly stated that arrest is inferred from an absence or reduction of pH3 positive cells.

According to the reviewer’s suggestion, we changed the following sentence on Page 12, line 187.

“To detect mitotic cells, we examined cell-cycle arrest by staining for phospho-histone H3 (pH3)¹⁸.”

↓

“Because cell-cycle arrest is inferred from reduction in the number of mitotic cells, staining for a mitotic marker, phospho-histone H3 (pH3), was performed¹⁸.”

Comment #3

Similarly in discussion of pH3 results, authors might want to consider that male and female germline are different. Myc downregulation could lead to arrest in female germlines but not male.

According to the reviewer’s suggestion, we added the following sentence on Page 13, line 197: “These results suggest that *Myc* downregulation did not cause cell-cycle arrest in male germline cells, but in female ones.”

Comment #4

*In the discussion the authors may wish to compare their observations with *myc* to the role of *wunen2* in determining pole cell elimination in the early embryos (Slaidina and Lehmann 2017).*

According to the reviewer's suggestion, we added the following sentences on page 18 line 270: “Our observations strongly suggest that elimination of germline cells mediated by *Myc* downregulation maintains the germline integrity. Recent study demonstrates that higher inheritance of germ plasm correlates well with higher survival probability of primordial germ cell (PGC) in embryos⁴⁹. This mechanism ensures selection of the PGCs with higher quantity of germ plasm. However, it is unlikely that this PGC selection depends on MDRG, because MDRG occurs in larvae, but not in embryos (Fig. 1h and g, and Fig. 2a and b).”